# Fast growth rate is associated with musculoskeletal biomechanical imbalance and dorsal cranial myopathy in broiler chickens

**Marconi Italo Lourenço-Silva**[1]*, **Anderson Hassell Norton III**[2], **Leonie Jacobs**[3]

**1** Department of Animal Production and Preventive Veterinary Medicine, School of Veterinary Medicine and Animal Sciences (FMVZ), São Paulo State University (UNESP), Botucatu, Brazil, **2** Department of Mathematics, Virginia Tech, Blacksburg, Virginia, United States of America, **3** School of Animal Sciences, Virginia Tech, Blacksburg, Virginia, United States of America

* marconi.italo@unesp.br

## Abstract

Dorsal cranial myopathy is a degenerative lesion that affects the *anterior Latissimus dorsi* muscle in broiler chickens, with an etiology that remains unknown. The objective was to investigate the influence of musculoskeletal biomechanical balance and gait on the prevalence of dorsal cranial myopathy in three broiler chicken strains with differing growth potential. Three-hundred and ninety-six broiler chickens from three genetic strains with differing growth potential (fast, intermediate, and slow, 132 birds/strain) were housed in 18 pens with 22 birds/pen. Five birds/pen (n = 30 birds/genetic strain) were randomly wing- or leg-banded to assess gait and musculoskeletal biomechanical balance (by calculating body angulation) at 1, 2, 3, and 3.7 kg weight sampling points. Dorsal cranial myopathy was assessed one day after birds reached final body weight. Gait and musculoskeletal balance were both negatively impacted by body weight in fast- and slow-growing strains but not in the intermediate-growing strain. Dorsal cranial myopathy was more prevalent in fast-growing broilers compared to other strains, with no case observed in the slow-growing strain. Impaired gait negatively affected musculoskeletal biomechanical balance and increased the prevalence of dorsal cranial myopathy. Our results suggest that genetic strain, musculoskeletal biomechanical imbalance, poor gait, and high body weight are all associated with the prevalence of dorsal cranial myopathy in broiler chickens. We successfully simplified a non-invasive body posture methodology to quantify the musculoskeletal biomechanical balance in broiler chickens.

## Introduction

Dorsal cranial myopathy (DCM) is characterized by muscular changes, including degenerative and multiphasic lesions that specifically affect the *anterior Latissimus dorsi* (ALD) muscle in chickens, a superficial, bilateral muscle located in the dorsal

**Data availability statement:** "Data underlying this manuscript are made accessible through the Virginia Tech Data Repository at https://doi.org/10.7294/29574800.v1.

**Funding:** LOURENÇO-SILVA, MI Coordination for the Improvement of Higher Education Personnel (CAPES) https://www.gov.br/capes. The funders had no role in study design, data collection and analysis, decision to publish, or preparation of the manuscript.

**Competing interests:** The authors have declared that no competing interests exist.

region of the wings, which functions as an abductor of the humerus and wings [1–4]. Macroscopically, the ALD muscle region affected by DCM shows yellow, odorless, gelatinous edema under the skin [5,6]. The muscle displays hemorrhagic lesions, pallor, increased thickness, firmer consistency, and adhesion to adjacent muscles [5,6]. Microscopically, this myopathy is characterized by multiphasic lesions with sparse muscle fibers, hyaline degeneration of fibers, necrotic fibers, extensive fibrous connective tissue proliferation, an increased number of satellite nuclei, and macrophages clearing the necrotic fibers [4,7].

Initially reported in slaughterhouse inspections in 2002, DCM has led to an increase in partial or total carcass condemnations [1]. This myopathy is typically detected only during processing, as it is difficult to identify in live birds. Small lesions that are only visible when the skin is ruptured may go unnoticed during routine *post-mortem* inspections. Thus, the reported DCM prevalence is considered an underestimation, at less than 1.5% [8]. However, in experimental settings, the prevalence can vary between 1% and 30% [1–3,9]. The myopathy has economic consequences because the affected area must be removed, which includes parts of the wing and breast muscle. Extensive lesions that impact the overall carcass appearance will lead to carcass condemnation in many countries (Ordinance No. 9,013 of 2017 from the Brazilian Ministry of Agriculture for food safety considerations; FSIS Directive No. 6100.3 of 2023 from the United States Department of Agriculture; and Regulation (EU 2019/627).

The etiology of DCM remains unknown. Observations suggest that this lesion primarily affects male broilers from fast-growing strains, which typically have good body condition, high slaughter weights, and no other visible issues or infections [7]. We theorize that fast-growing broiler strains are more prone to developing this myopathy than slower-growing strains, likely due to differences in the morphometric characteristics of their musculoskeletal systems [10,11]. In fast-growing broilers, the ALD muscle is atrophied due to their inability to fly. The large size of the pectoralis muscles may further impair ALD function, as these muscles are physically and mechanically unbalanced [1,2,11]. Additionally, fast-growing strains possess an underdeveloped musculoskeletal system, with tendons and bones that can be too weak to support their body weight [11–13]. Their bone structure often cannot keep pace with the rapid muscle growth, leading to an imbalance that overloads the musculoskeletal system and negatively affect their body posture, known as musculoskeletal biomechanical imbalance [10,11,13,14]. Slow-growing broilers do not experience these issues. Unlike fast-growing strains, they develop breast muscles more slowly [15], have longer bodies [16–18], and possess more developed leg muscles [15]. This may result in a stable musculoskeletal biomechanical balance as they age and no development of DCM. However, the stability of the musculoskeletal biomechanical balance as birds age, and the prevalence of DCM in intermediate- and slow-growing broilers have not been studied yet.

Fast-growing broilers with musculoskeletal biomechanical imbalance lean forward and downward into a tilted position [19]. This imbalance leads to discomfort, poor gait, valgus angular deformity, and spondylolisthesis [12,19,20]. Although slow-growing

broilers generally have stronger musculoskeletal systems than fast-growing strains [11], their musculoskeletal health and welfare may depend on their rearing system [21]. Slow-growing broilers raised in indoor systems exhibit reduced tibia strength and higher incidence of foot pad dermatitis and hock burns than those raised with outdoor access, due to lower stocking densities and increased opportunities for exercise [22,23]. In this study, we raised intermediate- and slow-growing strains in an indoor system, which may increase their susceptibility to musculoskeletal biomechanical imbalances.

Musculoskeletal biomechanical imbalance could be used as an *in-vivo* predictor for DCM in broilers. In response to poor balance, chickens raise their wings to maintain balance. This action contracts the atrophied ALD muscle, potentially worsening the DCM injury over time [5]. This theory is not confirmed by any studies. Musculoskeletal biomechanical balance can be evaluated by measuring body angulation through photogrammetry, a non-invasive method used in humans to assess equilibrium, center of gravity, balance oscillations, and body angulation [24]. This technique was adapted for broiler chickens [12,19]. In their approach, the authors considered the bird as a geometric figure (a spherical cap) and applied mathematical equations to calculate the center of gravity and the axis of symmetry from a photograph. Using these values, they quantified the body angulation of broilers when standing.

The potential association between musculoskeletal biomechanical imbalances and the prevalence of DCM in broiler chickens with different genetic strains has not yet been studied. Our objective was to determine how musculoskeletal biomechanical balance changes over time in three genetic strains with different growth potentials, its association with gait, and how these factors influence the prevalence of dorsal cranial myopathy. We predicted a negative association between body weight, gait score (worsened gait), and body angulation, with the latter representing a musculoskeletal biomechanical imbalance. We hypothesized a positive association between those metrics and the prevalence of dorsal cranial myopathy. We expected that fast-growing broilers exhibited poorer body angulation and gait compared to intermediate- and slow-growing strains, resulting in a higher prevalence of dorsal cranial myopathy.

## Materials and methods

The trial was carried out at Virginia Tech's Turkey Research Center. Virginia Tech's Institutional Animal Care and Use Committee approved the experimental protocol (number 23−015) and all procedures were performed following relevant guidelines and regulations. The trial consisted of a randomized block design of three genetic strains with six replicates.

### Birds, facilities, and management

Three-hundred and ninety-six one-day-old mixed-sex Cobb 500, Hubbard Redbro ColorYield, and Hubbard REDJAKi chicks were used (132 chicks for each strain). Chicks were obtained from a commercial hatchery where they were vaccinated for Marek's disease, followed by 6-h transportation to the research facility. The trial was carried out in a climate-controlled poultry barn with negative pressure ventilation. Birds were allocated in 18 pens (3 m²) with 22 birds each. Pens contained new pine shavings, one hanging galvanized tube feeder (SKU# CO30131, Hog Slat, Newton Grove, NC, USA), one water line with three nipple drinkers (Valco Industries, Inc., New Holland, PA, USA), and one A-frame plastic hut (29.9 cm L × 21.5 cm W × 21.4 cm H) as an enrichment item. House temperature was adjusted starting at 30°C on day 1 and reduced approximately 1.5°C every 2–3 days. The birds were maintained on an artificial lighting program of 24L:0D in the first 3 days due to heat lamps and 18L:6D until the end of the trial, with a light intensity of approximately 15 lx during light hours. Both feed and water were provided *ad libitum*. The commercial corn-soy diets were prepared according to the nutritional specifications for conventional broiler chickens and were separated into three rearing phases: starter (day 1–17 for Cobb 500 and day 1–23 for Hubbard Redbro ColorYield and Hubbard REDJAKi; 3000 kcal ME and 23% CP), grower (day 18–33 for Cobb 500 and day 24–56 for Hubbard Redbro ColorYield and Hubbard REDJAKi; 3100 kcal ME and 21.5% CP), and finisher (day 34–45 for Cobb 500, day 57–73 for Hubbard Redbro ColorYield, and day 57–85 for Hubbard REDJAKi; 3150 kcal ME and 20% CP) [25]. Each diet was provided for approximately 1/3rd of the production period, which was dependent on growth rates.

## Genetic strain

The study compared three broiler chicken strains with differing genetic growth potential (Fig 1). The Cobb 500 (Fig 1A) was categorized as a fast-growing strain with an average growth rate of 79 g/day at 45 days of age [26]. The Hubbard Redbro ColorYield (Fig 1B), which comes from a cross between the male New Coloryield and the female JA57Ki [27], was categorized as an intermediate-growing strain with an average growth rate of 35–42 g/day [27,28]. The Hubbard REDJAKi (Fig 1C), which comes from a cross between the female JA57Ki and the male RedBro [27], was categorized as a slow-growing strain with an average growth rate of 35 g/day at 85 days of age [27], Hubbard REDJAKi performance objectives, unpublished]. In our study, the Cobb 500 strain showed an average growth rate of 83 g/day, the Hubbard Redbro ColorYield strain showed an average growth rate of 50 g/day, and the Hubbard REDJAKi strain showed an average growth rate of 44 g/day.

## Measurements

Five wing- or leg-banded birds per pen (n = 30 birds/genetic strain) were randomly selected when each strain reached an average body weight of 1 kg (Table 1). Birds were gently marked on the head, back, and/or wings with a livestock marker (All-Weather Paintstik, LA-CO Industries, Inc., IL, USA). These markings were reapplied as necessary throughout the experiment. Birds were sampled for gait and angulation at four target body weights, based on average weights. In addition, dorsal cranial myopathy was assessed when birds reached 3.7 kg (Table 1).

   **Walking ability (gait).** A trained observer assessed the gait of all selected chickens when they reached 1-kg, 2-kg, 3-kg, and 3.7-kg body weights (Table 1). A six-point scale was applied to classify gait scores [29]: score 0, normal walking; score 1, the chicken moves fast, but a slight walking deficiency is observed; score 2, the chicken moves fast, but there

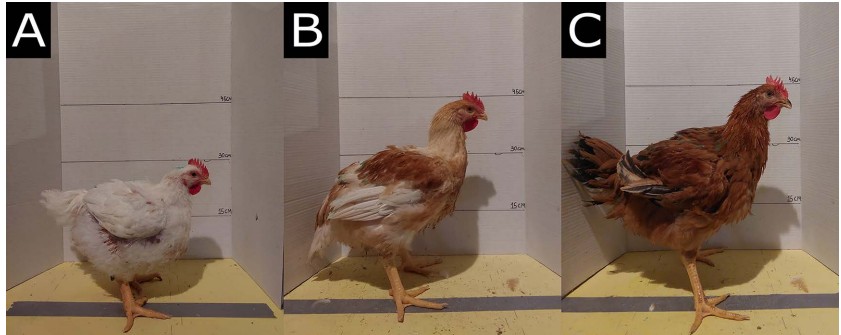

**Fig 1. Broiler chicken strains with differing genetic growth potential at 3 kg body weight. (A)** Cobb 500, **(B)** Hubbard Redbro ColorYield, and **(C)** Hubbard REDJAKi.

**Table 1. Age (days) and body weight (mean ± standard deviation, kg) of tested birds at each weight timepoint per genetic strain (with growth rate categorization fast, intermediate or slow).**

| Weight sampling timepoint | Cobb 500 (fast) | | Hubbard Redbro ColorYield (intermediate) | | Hubbard REDJAKi (slow) | |
|---|---|---|---|---|---|---|
| | Age (days) | Body weight (kg) | Age (days) | Body weight (kg) | Age (days) | Body weight (kg) |
| 1 kg | 20 | 0.98 ± 0.09 | 27 | 0.91 ± 0.09 | 30 | 1.02 ± 0.09 |
| 2 kg | 30 | 2.07 ± 0.23 | 43 | 1.93 ± 0.20 | 49 | 2.08 ± 0.25 |
| 3 kg | 38 | 3.04 ± 0.35 | 61 | 3.02 ± 0.37 | 67 | 3.01 ± 0.41 |
| 3.7 kg | 44 | 3.66 ± 0.39 | 67 | 3.56 ± 0.47 | 85 | 3.70 ± 0.50 |

is a significant walking deficiency; score 3, the chicken moves fast, but it presents an important walking deficiency; score 4, the chicken moves with a serious difficulty; and score 5, the chicken barely moves and often uses its wings during locomotion [29].

**Musculoskeletal biomechanical balance (body angulation).** After the gait assessment at each weight sampling timepoint, the birds were photographed to determine body angulation while standing in a lateral direction and assess the musculoskeletal biomechanical balance. Birds were placed on a table with a camera (Samsung SM-A725M/DS, Suwon-si, South Korea) positioned at a 30 cm distance. The camera was placed on a tripod at 30 cm height measured from the table top. We modified the method described by [12,19] to determine body angulation. Those researchers began with the assumption that they could fit each bird within a hemisphere of the same radius, r: "The sphere radius was measured, and this value was constant for all analyzed images, as the birds had similar weight and shape." [19, p. 374]. They fit the hemisphere so that this fixed radius—running from the center of the sphere, O, to either end of the hemisphere, A or B—was parallel to the line through the cloaca (C) and waddle (W; Fig 2). In other words, the length of segment OB is a radius of length r, and this segment is parallel to CW (see Fig 2). They seemed to assume (without explicitly stating) that O should lie along the line perpendicular to CW and passing through its midpoint, M. Thus, O and M would both lie along the line of symmetry (OH) of the hemisphere and the spherical cap—the part of the hemisphere that is below the line CW (Fig 2).

Next, they computed the location of the centroid of the spherical cap, E, along the line of symmetry, OH. To compute the distance, d, from O to E, the authors seem to have used Equation 1; where r is the length of radius OH, and h is the length of MH (the "internal shaft").

$$d = \frac{3}{4}\left[\frac{(2r-h)^2}{(3r-h)}\right]$$
(1)

$$d' = \frac{3}{4}\left[\frac{2(r-h)^2}{3(r-h)}\right] = \frac{(r-h)}{2}$$
(2)

The authors might have found the formula for d on Wolfram's Mathworld [30], which cites Harris and Stocker [31] as the original source. The original formula for d is Equation 1, but the formula published in both papers [12,19] is Equation 2, which simplifies to (r-h)/2. This would mean that the distance from O to E would be half the distance from O to M, which is incorrect. It appears that the authors used the correct formula for calculating d in their work but reported the incorrect formula. As r was assumed to be fixed for all chickens, the variable d (and thus the location of the centroid E) depends only on length h, which is approximately half of CW.

After computing distance d, the authors measured angle QEF as another important variable in assessing posture. This angle was created by first constructing a perpendicular line, EQ, from E to the ground. Then, by extending the radius OH into a line to the ground, at point F, they could measure angle QEF with a protractor. We note this angle is the same no matter where E is located along OH; and this angle is same as the angle between CW and a horizontal line.

We therefore modified the approach to obtain a more accurate measure of the same angle, which is less prone to measurement error. Namely, angle WMD is congruent to angle QEF and does not rely on first finding the position of the centroid, E, along OH. Thus, it provides for a more reliable measure of the same angle. Angle WMD is the same as angle QEF because (1) EQ is perpendicular to MD, and (2) EF is perpendicular to MW. The two angles are just positioned differently, with angle QEF rotated 90 degrees relative to angle WMD. Thus, both the variable E and the variable QEF could be replaced with simpler measures that depend only on the segment CW, the segment from the cloaca to the waddle: the distance d corresponds to half the length of MW; and the angle QEF is the same as the angle of tilt of MW, relative to horizontal. All calculations were made using Inkscape (version 1.2.2. Software Freedom Conservancy, NY, USA).

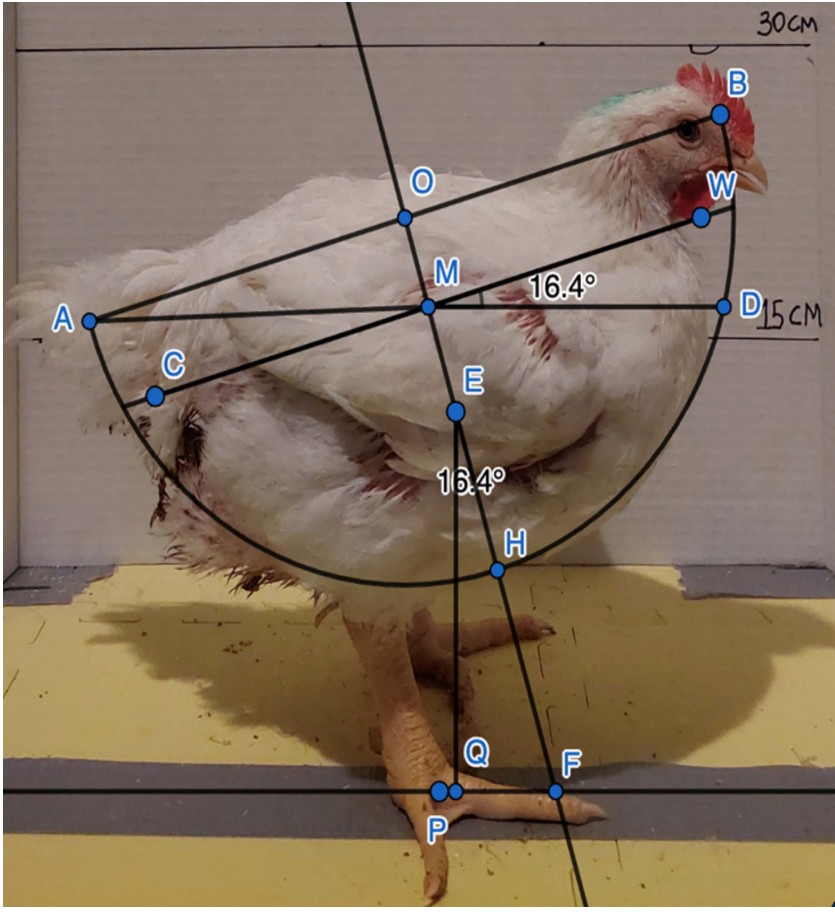

**Fig 2. Representation of the original method reported in [12,19] to calculate the body angulation and estimate the musculoskeletal biomechanic imbalance of broilers from a lateral view, plus the angle calculation as applied in the current study as shown as the angle of WMD.**

**Dorsal cranial myopathy.** The day after the birds reached 3.7 kg (Table 1), birds were euthanized via manual cervical dislocation by a trained researcher. The researcher secured the broiler's legs using one hand while providing support to the body with the upper leg. Subsequently, using two fingers, they firmly and swiftly grasped behind the skull. The head was then pulled down and turned against the knuckle of the first finger to elongate the neck and separate the skull from the atlas vertebra [32]. After euthanasia, the presence of dorsal cranial myopathy was macroscopically assessed in all animals. The anterior *Latissimus dorsi* (ALD) muscles were assessed for uni- or bilateral integrity and scored as 0: intact muscle, with no apparent macroscopic lesions; and 1: uni- or bilaterally affected muscle, with superficial hemorrhage, paleness, greenish colored exudate, or altered muscle color exhibiting necrosis and increased volume (Fig 3, adapted from [3]).

## Statistical analysis

Data were analyzed in SAS Studio 3.8 (SAS Institute Inc., Cary, NC, USA). The variance homogeneities were assessed by Levene's test (P = 0.108) and data residuals' normality was verified by the Shapiro-Wilk test (P = 0.166). Data distributions were assessed and verified by the Severity procedure. A generalized linear mixed model (GLIMMIX) was applied for gait score data using a gamma distribution; gait score was the response variable, genetic strain (n = 3), body weight (n = 4),

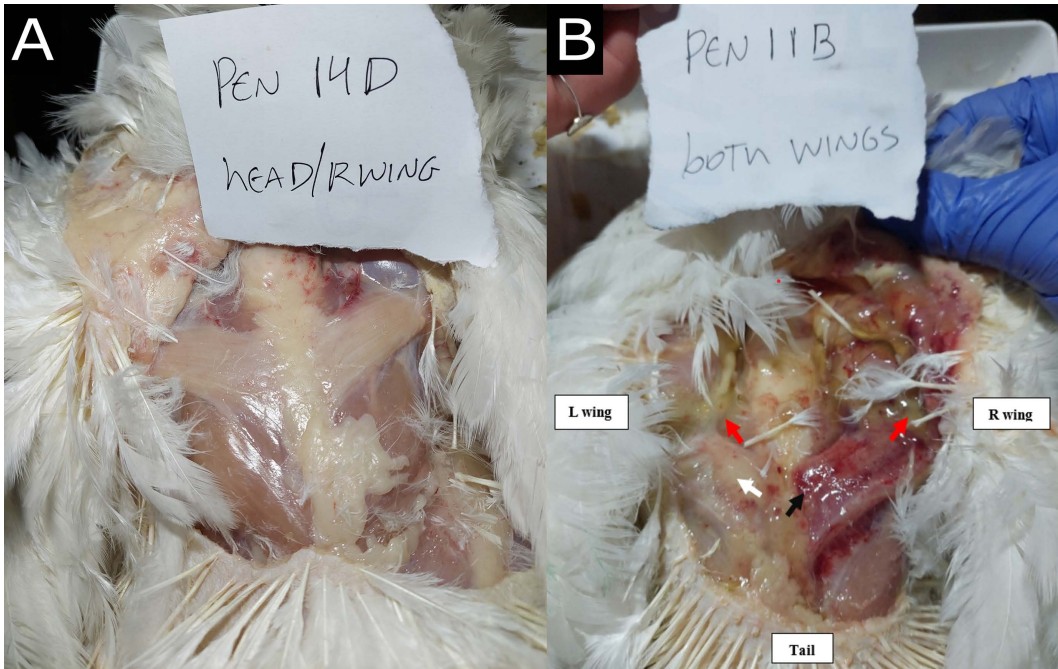

**Fig 3. Posterior view of fast-growing broiler chickens with or without bilateral dorsal cranial myopathy in the *anterior Latissimus dorsi* muscle at 45 days of age. (A)** Score 0: intact muscle, with no apparent macroscopic lesions, **(B)** Score 1: uni- or bilaterally affected muscle, with superficial hemorrhage (black arrow), paleness (white arrow), greenish colored exudate (red arrows) or altered muscle color exhibiting necrosis and increased volume.

and their interactions (n = 12) were fixed effects, and pen and bird ID were random effects. Body angulation data were subjected to ANOVA using a GLIMMIX, followed by Tukey's multiple comparison test and assigned significance when $P < 0.05$ with genetic strain (n = 3), body weight (n = 4), and their interaction (n = 12) as fixed effects and pen and bird ID as random effects. Dorsal cranial myopathy data were subjected to GLIMMIX using a gamma distribution with genetic strain (n = 3) as a fixed effect and pen as a random effect. Correlations between gait and body angulation, and dorsal cranial myopathy and body angulation at 3.7 kg were assessed using Spearman's correlation analysis with the CORR procedure, considering a significance of $P < 0.05$.

## Results

### Walking ability (gait)

An interaction between genetic strain and body weight for gait was found ($F_{6,104} = 5.54$, $P < 0.001$, [Fig 4]). Fast-growing broiler chickens showed better gait at 1 kg compared to 2 ($P < 0.001$), 3 ($P < 0.001$), and 3.7 kg ($P < 0.001$), and at 2 kg compared to 3 ($P = 0.004$) and 3.7 kg ($P < 0.001$). Gait did not differ at 3 and 3.7 kg ($P = 0.596$). Intermediate-growing broilers showed better gait than the fast-growing strain at 2 kg ($P = 0.005$), 3 kg ($P < 0.001$), and 3.7 kg ($P < 0.001$) but did not differ from the fast- ($P = 0.999$) or slow-growing strains ($P = 0.853$) at 1 kg, or from the slow-growing strain at 2 ($P = 1.000$), 3 ($P = 0.550$), and 3.7 kg ($P = 0.135$). Body weight did not impact gait for the intermediate-growing strain ($P > 0.772$). The slow-growing strain showed better gait than the fast-growing birds at 2 kg ($P = 0.001$) and 3 kg ($P = 0.023$) but did not differ at 1 ($P = 0.974$) and 3.7 kg ($P = 0.085$). Slow-growing chickens at 1 and 2 kg showed better gait than at 3.7 kg ($P < 0.001$), but gait did not differ at 1, 2, and 3 kg ($P > 0.089$) or when comparing 3 to 3.7 kg ($P = 0.497$).

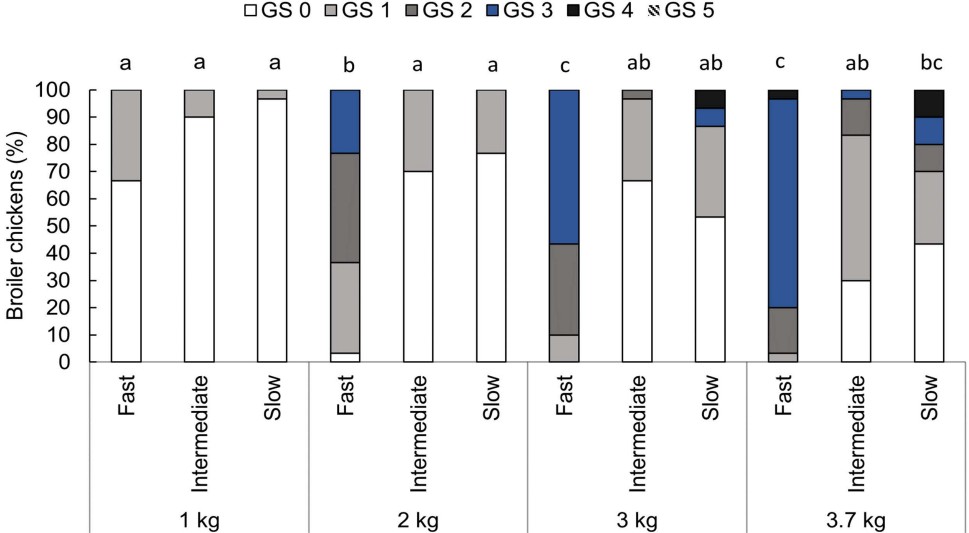

**Fig 4. Proportion (%) of broiler chickens categorized with each gait score (GS 0-5) by genetic strain (fast, intermediate, and slow) at weight points of 1, 2, 3, and 3.7 kg, n = 360 observations in 90 birds.** GS 0: normal walking, GS 1: the chicken moves fast, but a slight walking deficiency is observed, GS 2: the chicken moves fast, but there is a significant walking deficiency, GS 3: the chicken moves fast, but it presents an important walking deficiency, GS 4: the chicken moves with a serious difficulty, and GS 5: the chicken barely moves and often uses its wings during locomotion [29]. Bars with an uncommon superscript ($^{a-c}$) differed at P < 0.001.

## Musculoskeletal biomechanical balance (body angulation)

An interaction between genetic strain and body weight for body angulation was found ($F_{6,257}$ = 2.52, P = 0.022, Fig 5). Fast-growing broiler chickens showed a smaller angle than intermediate- and slow-growing strains at 1 kg (P < 0.001), 2 kg (P < 0.047), and 3.7 kg (P < 0.002). Fast-growing broilers at 3 kg showed a smaller angle than the intermediate-growing broilers (P < 0.001), but their body angle did not differ from the slow-growing broilers at that weight point (P = 0.981). The intermediate-growing strain showed a greater angle than the slow-growing strain at 3 kg (P < 0.001) but did not differ at 1 kg (P = 0.289), 2 (P = 0.555), and 3.7 kg (P = 0.871).

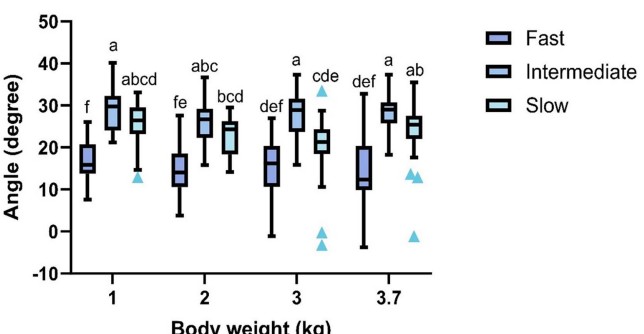

**Fig 5. Tukey box plots for angulation (degree) for broiler chickens from fast-, intermediate-, and slow-growing strains at weight points of 1, 2, 3, and 3.7 kg, n = 360 observations.** Boxes with an uncommon superscript ($^{a-f}$) differed at P < 0.001. Outliers are plotted as blue triangles.

## Dorsal cranial myopathy

Dorsal cranial myopathy prevalence differed between strains ($F_{2,72}$=4.61, P=0.013, Fig 6). Fast-growing broiler chickens showed a higher prevalence than intermediate- (P=0.043) and slow-growing (P=0.020) chickens. Dorsal cranial myopathy prevalences did not differ between intermediate- and slow-growing chickens (P=0.943).

## Correlation between the measures

A strong negative correlation between gait and body angulation was found (r=−0.526, P<0.001, n=360 observations), indicating that when gait scores were high (worse gait), the chickens' postural angles (musculoskeletal biomechanical balance) were low. A negative correlation between body angulation at 3.7-kg and dorsal cranial myopathy was found (r=−0.248, P=0.018, n=90 birds), indicating that when the chickens' postural angles were low, dorsal cranial myopathy prevalences were high. A positive correlation between gait score at 3.7-kg and dorsal cranial myopathy was found (r=0.227, P=0.031, n=90 birds), indicating when the gait scores at 3.7-kg body weights were high (poor gait), dorsal cranial myopathy prevalences were high.

## Discussion

This study investigated the musculoskeletal biomechanical imbalance in three genetic broiler chicken strains with fast-, intermediate-, and slow-growth potential. We evaluated the development of gait abnormalities and musculoskeletal biomechanical imbalance (body angulation) as the broilers gained weight, examined how these variables influenced the prevalence of DCM, and analyzed the association between these factors. We found that body weight negatively impacted gait and musculoskeletal biomechanical balance in both fast- and slow-growing strains but did not affect the intermediate-growing strain, which partially aligns with our hypothesis. Fast-growing broilers exhibited a higher prevalence of DCM compared to other strains, with no case observed in the slow-growing strain, which aligns with our hypothesis. Correlations between measures revealed that impaired gait is negatively associated with body angulation, meaning as gait scores increased (poor gait), body angulations decreased (poor musculoskeletal biomechanical balance). Furthermore, a

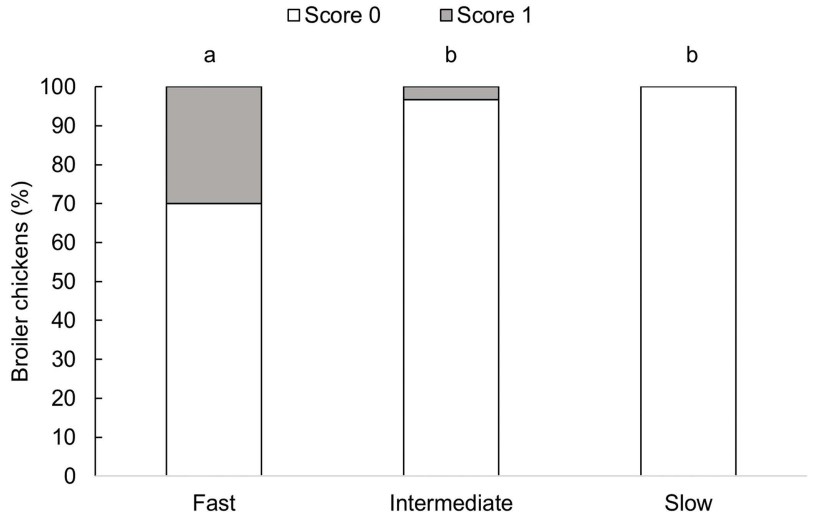

**Fig 6. Proportion (%) of broiler chickens with dorsal cranial myopathy by genetic strain (fast-growing at 45 days, intermediate-growing at 73 days, and slow-growing birds at 86 days of age), n=90 birds.** Score 0: intact muscle, with no apparent macroscopic lesions; Score 1: uni- or bilaterally affected muscle, with superficial hemorrhage, paleness, and gelatinous surroundings or altered color exhibiting necrosis and increased volume (adapted from [3]). Bars with an uncommon superscript ([a-b]) differed at P<0.05.

negative correlation was observed between body angulation and the prevalence of DCM, suggesting that poor musculo-skeletal biomechanical balance is associated with a higher prevalence of the condition, thereby supporting our hypothesis.

Our findings for gait in fast-growing broilers consistently align with prior studies, demonstrating that these animals are more likely to experience impaired gait as they reach slaughter weight [33–36]. Fast-growing broilers exhibit an awkward gait characterized by slow walking speed, a wide support base (pelvis), and large lateral motions [37]. Due to their anatomical characteristics, fast-growing broilers experience a high energetic cost of swinging their limbs during the final growth period [37,38], which leads to decreased activity levels [3,36,38,39]. This reduction in locomotor activity in fast-growing broilers then increases their susceptibility to develop leg disorders [40]. This explains the poor gait recorded in fast-growing birds at 3 and 3.7 kg body weights in the current study. It contrasts with the gait observed in intermediate-growing broilers, which exhibited better gait throughout the entire rearing period, regardless of body weight. The higher leg yield and lower abdominal fat composition in intermediate-growing broilers compared to fast-growing broilers [15,16,41] likely contributed to this difference, resulting in clinically better gait in the former.

Contrary to our hypothesis, slow-growing birds did not show better gait compared to intermediate-growing birds, and only showed improved gait compared to fast-growing birds at 2 kg and 3 kg body weights. We expected good gait, as their slow growth would result in improved muscle development. Still, slow-growing broilers exhibited worsened gait as they gained weight to the extent that their gait scores did not differ from the gait of fast-growing broilers at 3.7 kg. The poor gait at 3.7 kg could be linked to the housing conditions and the diets used in this study that exceeded the recommended metabolizable energy (kcal/kg) requirements for slow-growing broilers [42]. All strains were raised in pens in a poultry barn without outdoor access, relatively high stocking density, and received the same commercial diet to ensure consis-tent experimental conditions across all treatments. Slow-growing broiler strains are more commonly used in extensive housing systems with outdoor access [22], and have less abdominal fat when raised with outdoor access than indoors only [43]. Outdoor access improves gait scores, likely due to increased activity and exercise [44]. The lack of access to the outdoors, little space allowance in pens, and thus few opportunities to exercise may have contributed to poor gait in slow-growing birds. Additionally, the *ad libitum* diets in this study were formulated for fast-growing broilers [25], providing 120–240 kcal/kg more than required by slower-growing strains [42]. Combined with the limited space, this led to increased growth rates in slow and intermediate strains. The intermediate-growing strain showed an average growth rate of 49.5 g/day rather than 42 g/day [27,28], an 18% increase, and the slow-growing strain showed an average growth rate of 43.6 g/day rather than 35.0 g/day [27, Hubbard REDJAki performance objectives, unpublished], a 24.5% increase. The increased growth rate likely contributed to the poor gait at 3.7 kg. We speculate that the commercial diets and indoor environment for slow-growing broilers caused a nutrient imbalance and metabolic alterations (i.e., fat metabolism disorders) by affecting nutrient deposition and energy utilization [45]. In alignment, we observed that some slow-growing broilers developed leg disorders other than abnormal gait. An exploratory necropsy on two affected chickens revealed high abdominal fat depo-sition and femoral head separation, a degenerative problem typically found in fast-growing strains, where the growth plate of the proximal femur separates from its articular cartilage [46]. The high caloric intake impacts cartilage homeostasis and promotes the downregulation of basic fibroblast growth factor, increasing the risk of developing femoral head separation and contributing to the prevalence of lameness in our slow-growing broilers [47,48].

In the current study, age affected the musculoskeletal biomechanical balance of broilers differently depending on genetic strain. Fast- and slow-growing broilers showed reduced body angulation as they became heavier and older, but this change did not occur in intermediate-growing broilers, partially agreeing with our hypothesis. These results align with the walking ability results, evidenced by a strong negative correlation between gait score and body angulation. This indi-cates that when gait scores were high (poor gait), the chickens' postural angles were low (tilted downwards).

The musculoskeletal biomechanical balance differed between genetic strains due to body conformation differences. In fast-growing strains, rapid muscle growth and exaggerated development of the *Pectoralis major* muscle shift the chicken' center of gravity by moving the center of mass cranio-dorsally in the last two weeks of growth [12,19,38]. This shift causes

these chickens to lean forward, changing their posture, and increasing the load on their skeleton with increasing age [12,49]. The load directly impacts locomotion, increasing limb muscle stresses while standing and moving [38], which likely contributed to poor gait at 3 and 3.7 kg weight points. The intermediate-growing strain has smaller breast muscles and more developed leg muscles compared to fast-growing strains [15]. In contrast with fast-growing broilers, the slow development of the *Pectoralis major* muscle [15,18,22] and their longer body [16–18] allow these birds to maintain their center of gravity and center of mass, resulting in a consistent postural angle with age, thus an optimal musculoskeletal biomechanical balance. Our findings confirm that genetic selection for intermediate growth (or relaxed selection for growth) positively impacts the musculoskeletal biomechanical balance in broiler chickens.

However, this effect is only partially seen in the slow-growing strain, where imbalance did increase with age, likely due to housing conditions, diets, and subsequent leg disorders. This indicates that the musculoskeletal biomechanical balance is not only influenced by genetic growth potential and body conformation, but also by management and housing conditions. Our results suggest that intermediate-growing strains offer a better balance between growth rate and leg health compared to slow-growing strains when fed diets as in this study. We recommend further studies comparing performance and economic parameters of intermediate- and slow-growing broilers raised in conventional production systems to confirm our findings.

Fast-growing broilers showed a higher prevalence of DCM (9/30) compared to intermediate-growing broilers (1/30) and the slow-growing strain (0/30). Low postural angles were correlated with high dorsal cranial myopathy scores, and high gait scores (poor gait) were correlated with high dorsal cranial myopathy scores. Our findings indicate that poor balance, poor gait, and dorsal cranial myopathy are all more prevalent in fast-growing broilers at 3.7 kg live weight than the other strains, with the three conditions being interconnected.

We theorize that the heavy breast muscle and shorter bodies and keel bones in fast-growing broilers worsens their musculoskeletal biomechanical balance. This condition demands wing raising to maintain balance and locomote, as the center of gravity shifted forward with increased body (and specifically pectoral muscle) weight, ultimately resulting in dorsal cranial lesions. Excessive wing flapping is suggested as a cause of these lesions [5]. The ALD muscle is composed exclusively of red fibers, also known as type I fibers, which are more sensitive to hypoxia [50]. When contracting, the ALD muscle draws the wing caudally while flexing and elevating the humerus, thus regulating the movement of the humerus during the contraction of the *Pectoralis thoracicus* and supracoracoideus muscles [51]. When the wings move over the pectoral muscles, blood flow could be intermittently interrupted, leading to focal injury from reduced blood flow (ischemia) followed by restoration of blood flow (reperfusion), which leads to reactive oxygen species forming locally, damaging the ALD cells [2,50]. Additionally, normal ALD muscles will exhibit early degenerative primary microscopic lesions, which are less severe compared to lesions in muscles affected by DCM [2,6]. These microscopic lesions can develop repeatedly and lead to polyphasic late lesions (due to the self-perpetuation of the lesion by subsequent damage) [2,4]. This repeated damage is likely caused by excessive wing raising and flapping to maintain or regain balance.

In our previous work with fast-growing broilers, the provision of step platforms, straw bales, and moving laser lights resulted in a lower prevalence of DCM compared to those without enrichment [3]. We theorized that this was due to increased opportunities for exercise, that subsequently led to an improvement in musculoskeletal biomechanical balance [3]. This now seems to align with the current findings.

We modified and simplified the method described in [12,19] to measure body angulation, to quantify the musculoskeletal biomechanical balance in broiler chickens. The angle measured in the current study (WMD) has the same measure found using the methods described in [12,19]. By using the Inkscape software to measure the angle, we removed the time-consuming limitation reported in the previous studies [12,19]. Furthermore, we found that the distance from P to Q shown in Fig 2 could provide another valid measure for posture angulation. For instance, P is the point at the center of the footpad, and Q is the point on the ground directly below the centroid, E, whose position would be calculated by Equation 1. So, PQ would measure how far the centroid, E, lies in front of, or behind, the center of the footpad.

In the current study, we collected the chickens from their home pens to take the photos, rather than taking the photos in the home pen as reported in [12,19]. Anecdotally, we observed that some chickens showed vigilance behaviors (i.e., head up alert and looking around), which could impact the angle calculation and affect the body posture measurement. To avoid that, we allowed the chicken to investigate the area (habituate) and return to their natural head position before taking the photo.

In conclusion, our data show that genetic strain, musculoskeletal biomechanical imbalance, poor gait, and high body weight are associated with the prevalence of dorsal cranial myopathy in broiler chickens. Our results indicate that fast-growing broilers are more susceptible to developing this condition compared to intermediate- and slow-growing broilers. Although slow-growing broilers exhibited poor balance and gait at 3.7 kg, possibly due to housing without outdoor access and commercial fast-growing diets, dorsal cranial myopathy was not observed. This suggests that body conformation and the exaggerated development of the *Pectoralis major* muscle in fast-growing broilers play a more important role in the occurrence of this condition. We recommend further studies that directly assess body conformation and breast muscle development in relation to the progression of these lesions as birds age. Based on our findings, the intermediate-growing strain appears to be the most suitable for improving chicken health while maintaining reasonable growth rate under conventional broiler production systems with a target slaughter weight of 3.7 kg. We successfully simplified the body posture method to quantify the musculoskeletal biomechanical balance in broiler chickens. However, it is important to underline that this method may be affected by the chicken's head position. Thus, we suggest giving the chicken time to explore its surroundings and settle into a natural head position prior to taking the picture. This method can be used as a cost-effective indicator for monitoring musculoskeletal imbalance, dorsal cranial myopathy, and leg health in commercial settings.

## Author contributions

**Conceptualization:** Marconi Italo Lourenço da Silva, Leonie Jacobs.

**Data curation:** Marconi Italo Lourenço da Silva, Leonie Jacobs.

**Formal analysis:** Marconi Italo Lourenço da Silva, Leonie Jacobs.

**Investigation:** Marconi Italo Lourenço da Silva, Leonie Jacobs.

**Methodology:** Marconi Italo Lourenço da Silva, Anderson Hassell Norton III, Leonie Jacobs.

**Project administration:** Leonie Jacobs.

**Resources:** Leonie Jacobs.

**Supervision:** Leonie Jacobs.

**Validation:** Anderson Hassell Norton III, Leonie Jacobs.

**Writing – original draft:** Marconi Italo Lourenço da Silva.

**Writing – review & editing:** Anderson Hassell Norton III, Leonie Jacobs.

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
