## [Decision Letter · Decision Letter 0]

10 Jul 2025

Dear Dr. Lourenço da Silva,

Thank you for submitting your manuscript to PLOS ONE. After careful consideration, we feel that it has merit but does not fully meet PLOS ONE’s publication criteria as it currently stands. Therefore, we invite you to submit a revised version of the manuscript that addresses the points raised during the review process.

The manuscript addresses an important welfare-related condition in broiler chickens and is generally well-written. Both reviewers found the topic relevant and the methodology acceptable. However, clarification is needed on the feeding regimes, the term “hypercaloric diet,” and citation issues. Missing supplementary files should be made available. Expanding the discussion on practical applications and potential histological support is recommended. I recommend major revision before acceptance.

We look forward to receiving your revised manuscript.

Kind regards,

Arda Yildirim, Ph.D.

Academic Editor

PLOS ONE

3. Please upload a copy of Supporting Information Figure/Table/etc. which you refer to in your text on page 25.

Additional Editor Comments:

Dear Authors, Thank you for your submission. The reviewers found your study to be original and relevant, with potential value for both poultry science and broader animal welfare discussions. However, they also raised important points regarding the need for additional microscopic or molecular support, clarification of certain terms and procedures, and improved accessibility of supplementary materials. In light of these comments, a major revision is required before a final decision can be made. We encourage you to carefully address all reviewer suggestions to enhance the clarity and scientific rigor of your manuscript. Best regards, Arda Yıldırım

Reviewers' comments:

Reviewer's Responses to Questions

**Comments to the Author**

1. Is the manuscript technically sound, and do the data support the conclusions?

Reviewer #1: Yes

Reviewer #2: Yes

2. Has the statistical analysis been performed appropriately and rigorously?

Reviewer #1: Yes

Reviewer #2: Yes

3. Have the authors made all data underlying the findings in their manuscript fully available?

Reviewer #1: Yes

Reviewer #2: No

4. Is the manuscript presented in an intelligible fashion and written in standard English?

Reviewer #1: Yes

Reviewer #2: Yes

Reviewer #1: In this manuscript, three different strains of broiler chicken were compared in terms of gait, posture and myopathies in the M. latissimus dorsi. The manuscript is very well written, the contents can well be understood also by readers not that much familiar with poultry science. The contents are of high relevance demonstrating that high and disproportionate growth is at the costs of animal welfare. Research on myopathies in broiler chickens is largely focused on the pectoralis muscle as the most important muscle in terms of the meat product and its high share of total muscle mass. However, other muscles balancing the body posture must also be considered, and the non-invasive photogrammetric method described herein provides a valuable tool for assessing and predicting imbalances in the musculo-skeletal system.

Since PlosOne is targeting a broad scientific community, some poulty commons should be explained to non-expert readers: (1) the different times on the different diets (starter, grower, and finisher) were assumingly related to the growth velocity of the strains. However, it would be interesting to get to know whether the age at shifting to the next day was based on reaching a certain body weight and how the time of changing the diets might be related to some portion of the adult weight the animals would theoretically reach. (2) Some conclusions or suggestion would be helpful on how to implement the findings about genetic strains into chicken meat production systems aiming to improve health and welfare of the birds. A hypercaloric diet was mentioned in Line 366, but this might need some discussion in terms of feed efficiency. What losses and what gains would be encountered if using slower growing strains? Would a change in feeding or in final slaughter weight help? Is the posture assessment potentially applicable under field conditions?

Line 33: Correct typo in “Five bids/pen”.

Line 366: some explanation about the term “hypercaloric diet” should be provided (how much over needs, why used…).

Reviewer #2: This is an interesting study that investigates dorsal cranial myopathy (DCM) through the lens of musculoskeletal biomechanical balance and gait analysis.

Major Comments:

The study would be significantly strengthened by the inclusion of microscopic findings or molecular/biomarker data to support the macroscopic observations. As it stands, the conclusions are based solely on gross morphological assessments, which limits the depth of pathological characterization.

I was unable to access the supplementary materials submitted with the manuscript. Please ensure that these documents are available and accessible for review.

Minor Comments:

Line 188 – (P.374): It is unclear what this reference (P.374) pertains to. Please clarify or correct the citation.

**Do you want your identity to be public for this peer review?** For information about this choice, including consent withdrawal, please see our Privacy Policy

Reviewer #1: No

Reviewer #2: No

---

## [Author Response · Author response to Decision Letter 1]

11 Aug 2025

A rebuttal letter that responds to each point raised by the academic editor and reviewer(s) was uploaded as a separate file labeled 'Response to Reviewers'.

---

## [Decision Letter · Decision Letter 1]

3 Sep 2025

Fast growth rate is associated with musculoskeletal biomechanical imbalance and dorsal cranial myopathy in broiler chickens

PONE-D-25-26158R1

Dear Dr. Lourenço da Silva,

We’re pleased to inform you that your manuscript has been judged scientifically suitable for publication and will be formally accepted for publication once it meets all outstanding technical requirements.

Kind regards,

Arda Yildirim, Ph.D.

Academic Editor

PLOS ONE

Additional Editor Comments (optional):

Dear Authors,

Thank you for your revised submission. The revisions have satisfactorily addressed the majority of the reviewers' comments and concerns. I am pleased to inform you that your manuscript is now recommended for acceptance and publication. Regards, Arda Yıldırım

Reviewer #1:

Reviewer #2:

Reviewers' comments:

Reviewer's Responses to Questions

**Comments to the Author**

Reviewer #1: All comments have been addressed

Reviewer #2: All comments have been addressed

2. Is the manuscript technically sound, and do the data support the conclusions?

Reviewer #1: Yes

Reviewer #2: Yes

3. Has the statistical analysis been performed appropriately and rigorously?

Reviewer #1: Yes

Reviewer #2: Yes

4. Have the authors made all data underlying the findings in their manuscript fully available?

Reviewer #1: Yes

Reviewer #2: Yes

5. Is the manuscript presented in an intelligible fashion and written in standard English?

Reviewer #1: Yes

Reviewer #2: Yes

Reviewer #1: The open questions were adequately addressed, thank you. Otherwise, I have no further comments to add.

Reviewer #2: (No Response)

**Do you want your identity to be public for this peer review?** For information about this choice, including consent withdrawal, please see our Privacy Policy

Reviewer #1: No

Reviewer #2: No

---

## [Editor Report · Acceptance letter]

PONE-D-25-26158R1

PLOS ONE

Dear Dr. Lourenço da Silva,

I'm pleased to inform you that your manuscript has been deemed suitable for publication in PLOS ONE. Congratulations! Your manuscript is now being handed over to our production team.

Kind regards,

on behalf of

Prof. Dr. Arda Yildirim

Academic Editor

PLOS ONE